# An Intelligent Approach to the Unit Nesting Problem of Coil Material

**Dezhong Qi [1,2], Wenguang Yang [1], Lu Ding [1], Yunzhi Wu [1], Chen Tian [1], Lifeng Yuan [1], Yuanfang Wang [3,*] and Zhigao Huang [3]**

[1]  Hubei Agricultural Machinery Engineering Research and Design Institute, Hubei University of Technology, Wuhan 430068, China; derek@hbut.edu.cn (D.Q.); y18571459158@outlook.com (W.Y.); dl15686902963@163.com (L.D.); 102100030@hbut.edu.cn (Y.W.); 15327330378@163.com (C.T.); y1549815@163.com (L.Y.)

[2]  College of Mechanical and Electronic Engineering, Shandong Agricultural University, Taian 271018, China

[3]  Railway Locomotive and Vehicle Institute, Wuhan Railway Vocational College of Technology, Wuhan 430205, China; hzg@wru.edu.cn

\*  Correspondence: yuanfangwang@wru.edu.cn

**Abstract:** With the popularization of small batch production, the main cutting method for sheet metal parts has changed. Laser cutting has become the main production method for coil material cutting. Developing an irregular part nesting method for the continuous cutting of coil material is urgent. Based on the coil material cutting process, this paper proposes an intelligent approach for the unit nesting problem of coil material. Firstly, a unit nesting model of coil material was constructed. Secondly, an intelligent approach using an improved marine predator algorithm was used to solve this model. In solving the model, the minimum nesting unit was continuously updated by changing the position, angle, and quantity of the nesting parts. Thirdly, the geometric characteristics of this minimum nesting unit were extracted. Finally, the nesting units for production were obtained using a single row or opposite row of the minimum nesting unit. The computational results and comparison prove that the presented approach is feasible and effective in improving material utilization, reducing production costs, and meeting the requirements of the production site.

**Keywords:** laser cutting; the minimum nesting unit; improved marine predator algorithm

## 1. Introduction

It is urgent to develop resource-saving intelligent manufacturing under the constraints of the resources and environment. There are many advantages to sheet metal parts such as their light weight, high strength, good conductivity, and low cost. They are used in many industries, e.g., automotive, electronic appliances, and medical devices. The primary cutting method for sheet metal parts is punching. Due to the specification of the process, punching has several disadvantages: (1) the partial layout scheme is so simple that the utilization rate is low; (2) the molds have to be replaced regularly because of its short lift, which leads to high costs; and (3) the section of punched parts is rough that needs to be finished. With the advancement of laser technology, it is preferrable to use laser cutting instead of punching for its following advantages: (1) the workpiece of arbitrary shape can be cut; (2) it does not require mold; and (3) the cutting surface is smooth. Most of the existing research has been applied to the punching process; therefore, this is an innovative study on coil material.

As a research hotspot, cutting stock problems are extensively addressed in automobile, glass, paper, clothing, and other industries. According to the dimensions of parts, the cutting problem can be classified as a one-dimensional (1D), two-dimensional (2D), or three-dimensional (3D) cutting problem. According to the shape of the parts, it can also be divided into rectangular and irregular part nesting problems. This study is applied to the

sheet metal of the coil cutting process, and the optimized nesting map of 2D irregular parts is required.

Currently, how to use the minimum quantities of resources to solve the cutting problem is one of the main research directions. Many researchers are drawn to the study of 2D cutting stock problems. For sheet cutting problems, Feng et al. [1] proposed a mathematical model of local fitness to measure the fitting degree of polygons. Gao et al. [2] preprocessed a combinatorial operation of parts to improve the performance of a 2D irregularly shaped part layout. Rao et al. [3] proposed the concept of a collision-free region (CFR) to solve the problem of the beyond boundary constraint. Ji et al. [4] proposed a new algorithm of two-staged cutting patterns. Wu et al. [5] proposed a manufacturability-oriented rectangular part cutting stock method to consider the optimization objective of obtaining high material utilization and a short cutting path. İsmail [6] introduced a hybrid algorithm that combines the fruit fly optimization algorithm (FOA) with the bottom-left fill approach to solve the 2D strip packing problem. Nogueira et al. [7] presented a non-linear mathematical model and its linearization to represent the 2D cutting stock problem with usable leftovers (2D-CSPUL). Anand and Babu [8] introduced a novel approach that used heuristic and genetic algorithms for two-dimensional nesting. Ranga et al. [9] proposed a heuristic method based on the Jostle algorithm. Mellouli et al. [10] presented an innovative genetic algorithm (GA) for the multi-objective optimization of 2D nesting. Jiang et al. [11] proposed an uncertain optimization method that improves computational efficiency under the premise of considering the stiffness of the material. Prasad et al. [12] presented a novel computational strategy for collision detection to reduce the enormous computational time. Diyaley et al. [13] proposed a meta-heuristic algorithm, and it has an effective utilization ratio, nested height, and computational effort. For the cutting problem of coils, Qing et al. [14] divided the coil into segments with different lengths, and the segments were cut into rectangular pieces. Deng et al. [15] divided the coil into different strips, and then each strip was cut into desirable rectangles. The above method is suitable for rectangular parts, which will cause great waste for irregular parts. In addition, the emergence of new intelligent algorithms (marine predator algorithm) also provides a direction for the optimization of parts position sequence. Hung [16] used the marine predator algorithm to improve image quality. Sumit et al. [17] applied marine predators to engineering problems. Based on the analysis, this study proposes a unit nesting model for coils, optimized the marine predator algorithm, used the improved marine predator algorithm to solve the model, and designed a decoding method.

The rest of the paper is structured as follows: Section 1 proposes a mathematical model of coil material. Section 2 presents an intelligent approach to solving the mathematical model. The computational results and comparison are outlined in Section 3. The conclusions are summarized in Section 4.

## 2. Problem Model

At present, the main cutting methods are punching (Figure 1a) and laser cutting (Figure 1b). Based on the coil material cutting process in laser cutting, the approach for the unit nesting problem is proposed and the mathematical model is constructed in this section.

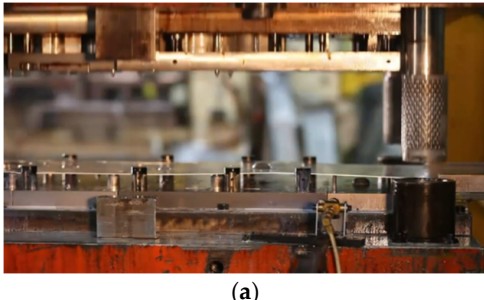
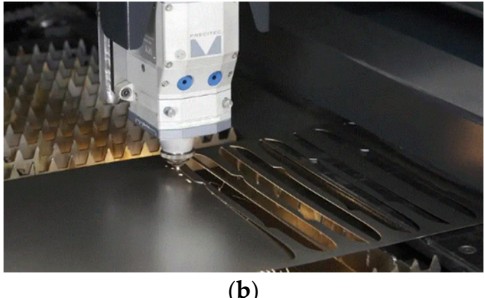

(**a**)　　　　　　　　　　　　　　　　　(**b**)

**Figure 1.** The method of cutting: (**a**) punching; (**b**) laser cutting.

*2.1. Problem Description*

The nesting problem of coil material is defined as follows: determination of a layout cut '*r*' different parts to achieve the maximum utilization (i.e., minimum length) of the coil with width (*W*). In the nesting process, the following constraint conditions need to be satisfied:

- The parts should be placed within the boundaries of the coil material;
- The part does not overlap with others;
- Each part meets the requirement of the quantity.

To achieve the maximum utilization, the rotation of the part should be fully considered. There are '*N*' parts, and the minimum rotation angle for each part is '*θ*', then the number of layout schemes is $N! \times \left(\frac{360}{\theta}\right)^N$. While the number of parts is increased and the minimum rotation angle for each part is decreased, the number of cutting patterns significantly increases under the same calculation conditions. Because the sheet layout algorithm studies the raw materials with a fixed length, it is not fully applicable to the coil. Thus, a new approach should be found.

*2.2. Conceptual Overview*

2.2.1. Unit

In the cutting problem of large batch parts, the complexity of nesting increases significantly with the number of parts. To address this issue, the concept of a "unit" is introduced. In Figure 2, a module and its external contour, called a unit, are combined by some parts. Figure 2b shows a unit that is composed of four parts according to a specific strategy, and different units can be obtained by changing the angles of parts.

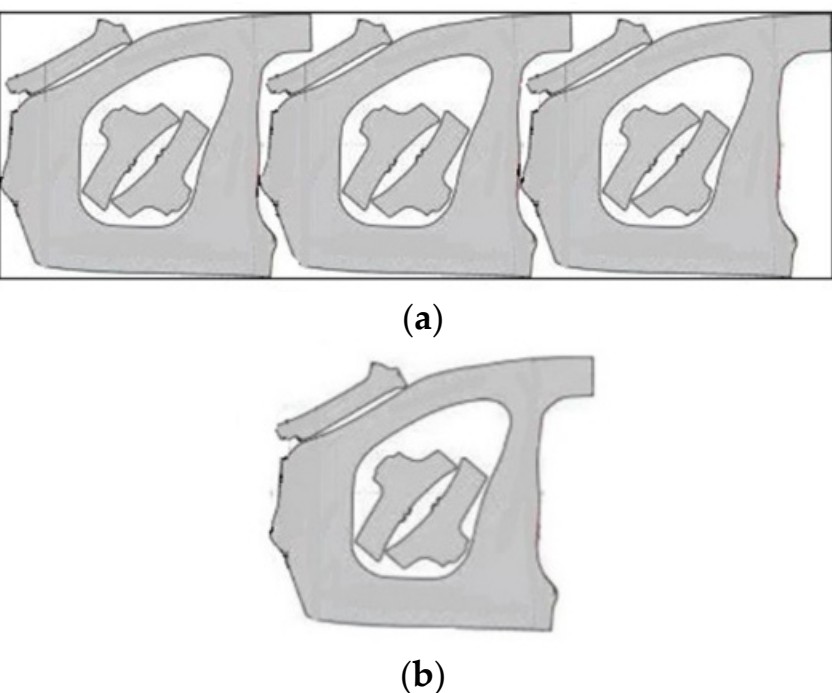

(a)

(b)

**Figure 2.** Diagram of unit: (**a**) diagram of cutting pattern; (**b**) minimum unit.

2.2.2. Unit Nesting Approach

The process of obtaining the cutting pattern by the repeated arrangement of units is called the unit nesting approach. The flowchart is shown in Figure 3.

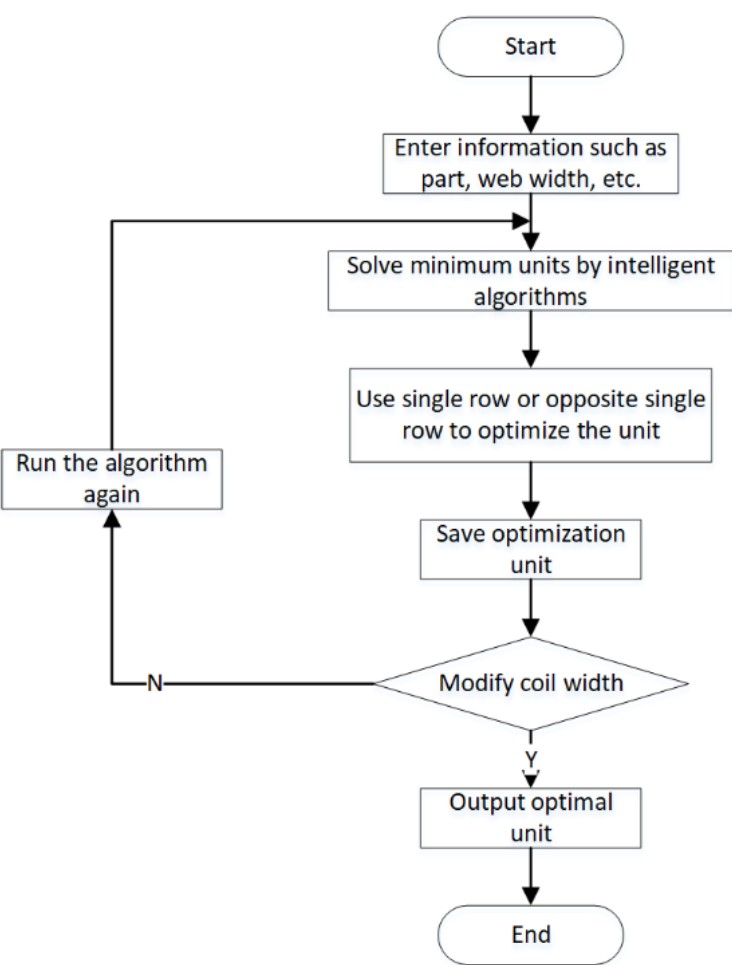

**Figure 3.** Flowchart of the unit nesting approach.

*2.3. Mathematical Model*

The variable definitions, used to establish a mathematical model for the irregular part nesting problem of coil material, are as shown in Table 1.

**Table 1.** Variable definitions.

| Variable | Variable Definitions |
|---|---|
| $W$ | The width of the coil |
| $D$ | Within the area of coil |
| $r$ | The quantity of part types |
| $N$ | The maximum quantity of parts in a unit |
| $M_i$ | The number of parts $i$ in a unit |
| $r_i$ | Within the area of part $i$ |
| $S_i$ | The area of parts $i$ |
| $\alpha_i$ | The angle of parts $i$ |
| $I$ | The placement order of parts |
| $L$ | The maximum length of a unit |
| $L'$ | The maximum length of two units after combination |
| $sl$ | The minimum distance of the unit movement without colliding ($sl = L' - L$) |

In Figure 4, '$L'$' represents the distance between the rightmost and leftmost points of a unit, and '$L''$' represents the distance between the leftmost and rightmost points of a combined unit.

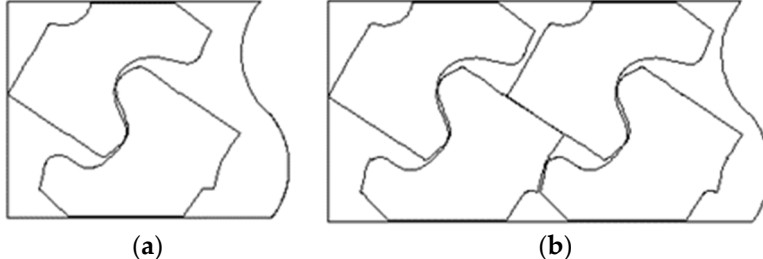

**Figure 4.** Illustration of partial variables: (**a**) illustration of '*L*'; (**b**) illustration of '*L*''.

In the nesting process of coil material, the position and orientation of parts are changed to achieve maximum utilization of the unit. The formula is shown as follows:

$$\max f(I, [\alpha_1, \alpha_2, \cdots, \alpha_N]) = \frac{\sum_{i=1}^{r} S_i \times M_i}{sl \times W},\tag{1}$$

$$\text{s.t. } r_i \in D,\tag{2}$$

$$r_i \cap r_j = \varnothing, i \neq j,\tag{3}$$

$$M_i \geq 1,\tag{4}$$

$$\sum_{i=1}^{r} M_i \leq N,\tag{5}$$

$$sl \leq 15{,}000.\tag{6}$$

In the above mathematical model, Equation (1) represents the optimization objective of the model. '*f*' is the utilization rate of the unit in the nesting layout, and it is influenced by the position and the orientation of parts. Equations (2) and (3), respectively, indicate that the parts should be placed within the boundaries of the coil material and do not overlap with others. Equations (4) and (5) represent the limitation of the number of parts in the unit to be met. Equation (6) represent that the maximum value of step length is 15,000.

## 3. Solution of the Model

The 2D nesting problem is a combinatorial problem. Because of the complexity of its solving space, the optimal solution is difficult to obtain. Therefore, it is called the NP complete problem. With the development of computers, an ideal solution can be found using intelligent algorithms in a short time, which is an important method for solving such problems. Thus, the section will be divided into a positioning strategy and sequencing strategy to introduce the solution.

### 3.1. Positioning Strategy

The positioning strategy is one of the key steps in the nesting problem. Currently, the common positioning strategies of the nesting problem include the bottom-left strategy (BL) [18], the bottom-left-fill algorithm (BLF) [19], the no-fit polygon algorithm (NFP) [20], and so on. These algorithms are simple to implement and have wide applications but are underutilized. To improve the situation, a hybrid algorithm of Tecnicas dé Optimização para o Posicionamento de Figuras Irregulares (TOPOS) and the lowest horizontal line (LH-TOPOS) for positioning is proposed.

### 3.1.1. NFP

NFP is an important geometric tool in nesting problems. By providing the position of the polygon, NFP can avoid a lot of repeated steps of intersection and collision, thus reducing the time complexity of the calculation. As shown in Figure 5, the concept of the no-fit polygon is defined as follows: two polygons are given, *A* and *B*. While *A* is fixed, reference point *P* is selected in *B*. *B* is sliding around the boundary of *A* to ensure that *A* and *B* are always tangent. The NFP$_{AB}$ is a closed path of the reference point *P* around a circle.

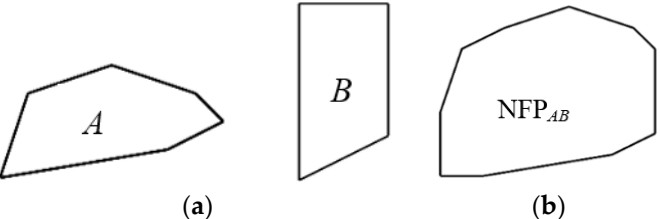

(**a**)                              (**b**)

**Figure 5.** Illustration of the NFP: (**a**) polygon *A*, *B*; (**b**) NFP$_{AB}$.

The geometric properties of NFP are as follows: if *P* is inside NFP$_{AB}$, *A* and *B* will overlap; if *P* is on the boundary of NFP$_{AB}$, *A* and *B* will touch without overlap; if *P* is outside NFP$_{AB}$, *A* and *B* will neither touch nor overlap.

### 3.1.2. Greedy Strategy

In the positioning strategy, a greedy strategy is introduced to obtain a higher utilization unit. It prioritizes the combination of parts with the next part. If the utilization rate of combined parts is greater, the combined parts are used to participate in the positioning strategy. The evaluation function is calculated as follows:

$$f = \frac{S_A + S_B}{S_{all}},$$

(7)

where $S_A$ is the area of part *A*, $S_B$ is the area of part *B*, and $S_{all}$ is the area of the circumscribed envelope rectangle of the combined part. If *f* is greater than the ratio of the area of the single part to its circumscribed envelope rectangle, it is considered an advantageous combination. Otherwise, the single part is considered advantageous.

### 3.1.3. TOPOS

When the feasible position of *B* relative to *A* is determined by NFP, the next step is to determine the optimal placement position. According to reference [21], the minimum length of envelope rectangle in criterion 2 is used as the basic evaluation criterion, and then the optimal position is selected by the lower horizontal line. Figure 6 shows the second principle of TOPOS.

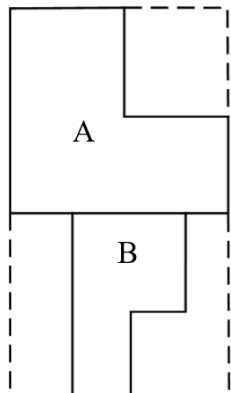

**Figure 6.** The second principle of TOPOS.

### 3.1.4. The Optimization of Unit

When the minimum unit is obtained, we need to optimize the minimum unit to obtain the unit for production. As shown in Figure 7, the utilization rate of unit b is higher than that of unit a, then b is the optimized unit for production.

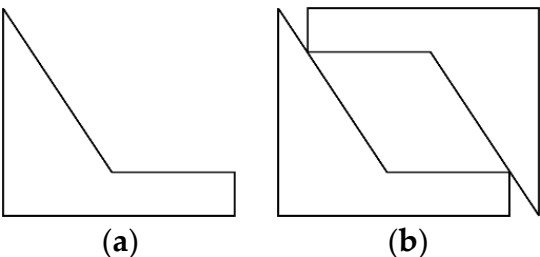

(a)   (b)

**Figure 7.** The optimization of the minimum unit: (**a**) the single row of the minimum unit; (**b**) the opposite row of the minimum unit.

### 3.1.5. LH-TOPOS

The flowchart of the LH-TOPOS is shown in Figure 8.

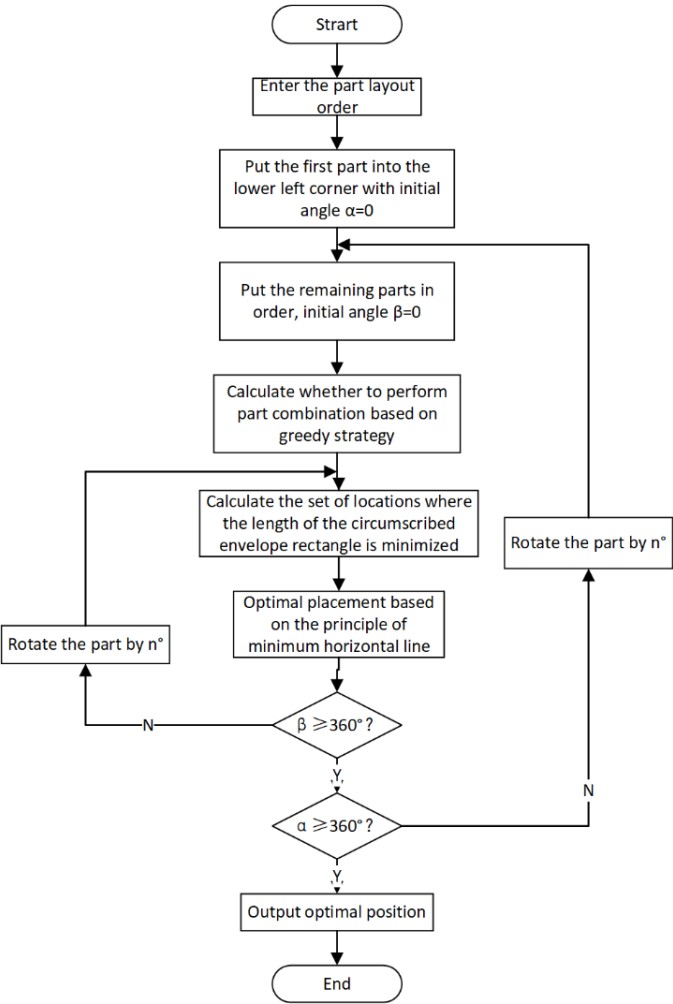

**Figure 8.** The flowchart of the LH-TOPOS.

### 3.2. Sequencing Strategy

When the positioning strategy is determined, only the influence of the nesting sequence of parts needs to be considered. Currently, many algorithms have been applied

to the sequencing strategy, such as the genetic algorithm (GA) [10], particle swarm optimization (PSO) [22], simulated annealing (SA) [23], ant colony optimization (ACO) [24], and so on. According to the references [25–27], the optimization ability of the marine predator algorithm (MPA) is significantly higher than that of PSO and GA, and it has been widely used in some fields. Thus, the improved marine predator algorithm is used as the sequencing strategy.

### 3.2.1. Encoding and Decode

The method of decimal coding is used to encode, and the part will be numbered $I = 0$, $1, \ldots, r$. The individual $I = \{I_1, I_2, \ldots, I_N\}$, $0 \leq I_i \leq r$, represent a solution to the nesting problem. For example, $\{2, 1, 0, 2, 0, 3\}$ represent the nesting map of the order of 2, 1, 2, 3.

The decoding method is LH-TOPOS described in the positioning strategy.

### 3.2.2. Initial Population

Similar to other algorithms, the MPA generates the initial population by random generation. Its generating formula is as follows:

$$X_{i,j} = X_{\min} + rand(X_{\max} - X_{\min}), \tag{8}$$

where $X_{\max}$ and $X_{\min}$ represent the upper and lower values of the variables, and *rand* represents a random number in the range [0, 1]. The nesting problem is a discrete problem, so it is necessary to discretize the initial population. In this paper, the data are discretized by the equidistant dispersion method. The $[X_{\min}, X_{\max}]$ is divided into r + 1 intervals on average, and the integer of $X_{i,j}$ is valued according to the interval of the data.

Based on Formula (9), the following matrix called *Prey* is generated:

$$Prey = \begin{bmatrix} X_{1,1} & X_{1,2} & \cdots & X_{1,d} \\ X_{2,1} & X_{2,2} & \cdots & X_{2,d} \\ X_{3,1} & X_{3,2} & \cdots & X_{3,d} \\ \vdots & \vdots & \vdots & \vdots \\ \vdots & \vdots & \vdots & \vdots \\ X_{n,1} & X_{n,2} & \cdots & X_{n,d} \end{bmatrix}_{n \times d}, \tag{9}$$

where *n* is the number of populations. *d* is the dimension of the nesting problem. $Prey_j = [X_{j,1}, X_{j,2}, \ldots, X_{j,n}]$ is a solution to the problem. Then, the optimal solution in *Prey* is replicated to obtain the matrix called Elite with the same dimension.

$$Elite = \begin{bmatrix} X_{1,1}^{Iter} & X_{1,2}^{Iter} & \cdots & X_{1,d}^{Iter} \\ X_{2,1}^{Iter} & X_{2,2}^{Iter} & \cdots & X_{2,d}^{Iter} \\ X_{3,1}^{Iter} & X_{3,2}^{Iter} & \cdots & X_{3,d}^{Iter} \\ \vdots & \vdots & \vdots & \vdots \\ \vdots & \vdots & \vdots & \vdots \\ X_{n,1}^{Iter} & X_{n,2}^{Iter} & \cdots & X_{n,d}^{Iter} \end{bmatrix}_{n \times d}, \tag{10}$$

where *Iter* is the current iteration.

### 3.2.3. Fitness Function

The fitness function is a parameter used to evaluate the quality of an individual and further guides the direction of evolution. The unit nesting algorithm has repeatability,

so the utilization rate of a unit is used as the fitness function in the cutting problem. Its expression is defined as follows:

$$\max f(I) = \frac{\sum\limits_{i=1}^{r} S_i \times N_i}{sl \times W},$$ (11)

3.2.4. Selection and Update

The process of selection and updating in MPA is divided into three stages based on different speed ratios. The optimal solution is obtained by changing the movement of the predator and prey at different stages.

Phase 1: The prey moves faster than the predators, and the algorithm is in the global search stage. The formula of prey is as follows:

$$While\ Iter < \frac{1}{3}Max\_Iter,$$

$$stepsize = R_B \otimes (Elite - R_B \otimes Prey_j),$$

$$Prey = Prey + P \cdot R \otimes stepsize,$$ (12)

where *Iter* represents the current iteration. *Max_Iter* represents the maximum iteration time. *stepsize* represents the moving step size in the stage. $R_B$ represents a random parameter based on Brownian motion. The notation $\otimes$ represents operator by term-by-term multiplication. $P = 0.5$. *R* represents a d-dimensional vector, and each value is a random number in the range [0, 1].

Phase 2: Both prey and predators move at the same speed. The population is divided into two parts, one of which is responsible for exploitation in the search space, and the other is responsible for exploration in the search space. Their formulas are as follows:

$$While\ \frac{1}{3}Max\_Iter < Iter < \frac{2}{3}Max\_Iter,$$

Prey (the first half of *Prey*):

$$stepsize = R_L \otimes (Elite - R_L \otimes Prey_j),$$

$$Prey_j = Prey_j + P \cdot R \otimes stepsize,$$ (13)

Predator (the second half of *Prey*):

$$stepsize = R_B \otimes (R_B \otimes Elite - Prey_j),$$

$$Prey_j = Elite + P \cdot CF \otimes stepsize,$$ (14)

where $R_L$ represents a random parameter based on Levy flight. *CF* represents an adaptive parameter of the predator's moving step, $CF = \left(1 - \frac{Iter}{Max\_Iter}\right)^{\left(2\frac{Iter}{Max\_Iter}\right)}$.

Phase 3: The prey moves lower than the predators. The formula for predators is as follows:

$$While\ Iter > \frac{2}{3}Max\_Iter,$$

$$stepsize = R_B \otimes (R_B \otimes Elite - Prey_j),$$

$$Prey_j = Elite + P \cdot CF \otimes stepsize, \tag{15}$$

The pseudo-code of population update is shown in Algorithm 1:

---
**Algorithm 1** Population update

---
1:  **while** Termination condition not met
2:      **if** $Iter < \frac{1}{3} Max\_Iter$
3:          Update Prey based on Formula (10)
4:      **else if** $\frac{1}{3} Max\_Iter < Iter < \frac{2}{3} Max\_Iter$
5:          pdate the first half of Prey based on Formula (11)
6:          Update the second half of Prey based on Formula (12)
7:      **else** $\frac{1}{3} Max\_Iter < Iter < \frac{2}{3} Max\_Iter$
8:          Update Prey based on Formula (13)
9:      **end if**
10: **end while**

---

### 3.2.5. Eddy Formation and FAD Effects

The eddy formation or fish aggregating device (FAD) effects are introduced in MPA to prevent the low efficiency and local optimum of the algorithm. In this case, the predator can take a longer jump to find an environment for other prey distributions. The formula of the FAD effect is as follows:

$$Prey_j = \begin{cases} Prey_j + CF[X_{\min} + R \otimes (X_{\max} - X_{\min})] \otimes U & if\ r \leq \text{FADs} \\ Prey_j + [\text{FADs}(1 - r) + r](Prey_{j1} - Prey_{j2}) & if\ r > \text{FADs}' \end{cases} \tag{16}$$

where $j1$ and $j2$ are random indexes of *Prey*, and $1 \leq j1, j2 \leq n$. $r$ is a random value from 0 to 1. FADs = 0.2. U represents a d-dimensional vector, and each value in U is generated by the following formula:

$$U = \begin{cases} 0 & if\ rand \leq FADs \\ 1 & if\ rand > FADs' \end{cases} \tag{17}$$

where *rand* is a random number between 0 and 1.

### 3.2.6. Marine Memory

This process is to update the *Elite*. For each individual in *Prey*, if the fitness value of $Prey_j$ is greater than the fitness value of $Elite_j$, the individual will be used to replace the corresponding individual in *Elite*.

### 3.2.7. Termination Criterion of Algorithm

There are two termination criteria of the algorithm, and the algorithm will be terminated when one of them is reached. One is to reach the preset maximum number of iterations ($T_{\max}$ = 1000), and the other is to obtain the same continuous optimal solution of 50 generations.

### 3.2.8. Improved Method

To further improve the ability of the algorithm, the algorithm will be improved in two aspects.

(1) In the initialization, the generation of the population is random, which will affect the convergence speed of the algorithm to a certain extent. Therefore, a combination of Tent mapping and random formation is used to generate the initial population, and the Tent mapping formula is as follows:

$$y_{t+1} = \begin{cases} \frac{y_t}{\beta} & 0 < y_t \leq \beta \\ \frac{1 - y_t}{1 - \beta} & \beta < y_t \leq 1 \end{cases}, \tag{18}$$

where $t$ represents the number of current iteration time, $T$ represents the maximum iteration time, and $\beta$ is a constant from 0 to 1. According to the reference [28], $T = 500$ and $\beta = 0.7$.

(2) In the population update stage, the MPA has different requirements at different stages. To improve the probability of successful predation, a nonlinear convergence factor $\alpha$ is introduced to adjust the moving step length of each stage to suit its different requirements. Then, *stepsize = stepsize $\times$ $\alpha$*. The convergence factor $\alpha$ decreases nonlinearly with the number of iteration increases. Early on, $\alpha$ decreases slowly to improve the global search capability. Later, $\alpha$ decreases rapidly to improve the local search capability. The rule for $\alpha$ is as follows:

$$\alpha = 1 - \left( \frac{t}{T_{\max}} \right)^{\lambda} \times (e + \mu) + \theta \times \kappa, \tag{19}$$

where $\theta$ is a random variable from 0 to 1, $t$ is the current iteration time, $T_{\max}$ is the maximum iteration time, and $e$ is Euler's Constant. According to the reference [29], $\lambda = 0.1$, $\mu = 1$, and $\kappa = 5$.

### 3.3. Algorithm Procedure

The specific steps of the unit nesting approach are as follows:

Step 1: Initialize the population;
Step 2: Calculate the initial fitness values by the LH-TOPOS, and obtain the Prey and Elite;
Step 3: Update the Prey according to different phases;
Step 4: Update the Prey and Elite by applying the FAD effect;
Step 5: Check if the termination condition is satisfied. If yes, output result; otherwise, return to Step 3.

The flowchart is shown in Figure 9:

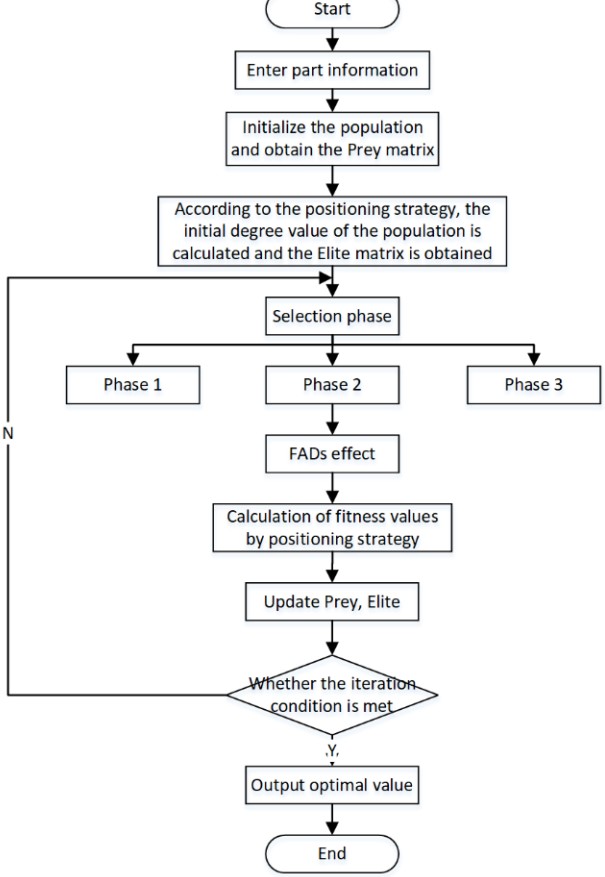

**Figure 9.** The improved marine algorithm.

## 4. Results

The practical data of an auto parts factory are used to verify the performance of the algorithm. Case 1 is the nesting of a single part, and case 2 is the nesting of many kinds of parts. We have experimented this on a system with two processors with 64 GB RAM, and each processor has 20 cores.

### 4.1. Case 1

In case 1, the nesting pattern of a single part is required. Table 2 is the part information to be produced. Figures 10–12 show the optimized nesting pattern and unit of part 1, part 2, and part 3, respectively. Table 3 is a comparison of the step length (the minimum distance of the unit movement without colliding) and the utilization.

**Table 2.** Single row parts information.

| Part Number | Legend | Size (mm × mm) | Coil Width (mm) |
|:---:|:---:|:---:|:---:|
| 1 | | 702 × 357 | 1260 |
| 2 | | 1973 × 466 | 1850 |
| 3 | | 897 × 326 | 1260 |

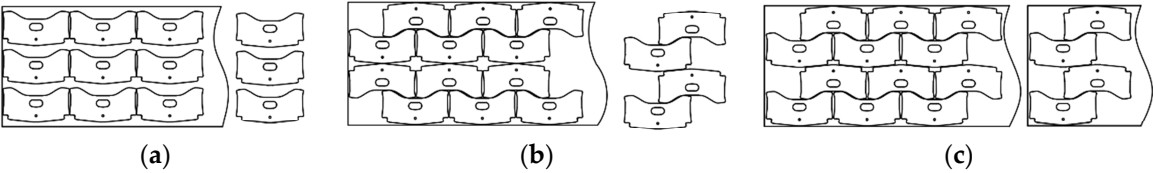

**Figure 10.** Part 1: (**a**) manual layout; (**b**) PSO; (**c**) unit nesting approach.

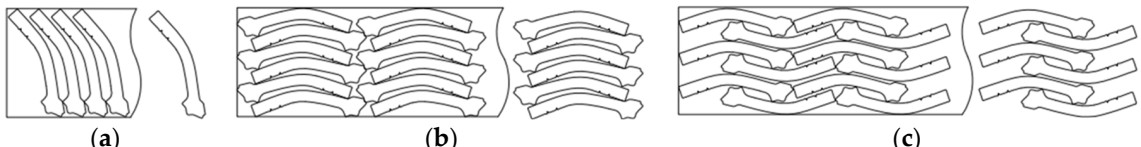

**Figure 11.** Part 2: (**a**) manual layout; (**b**) PSO; (**c**) unit nesting approach.

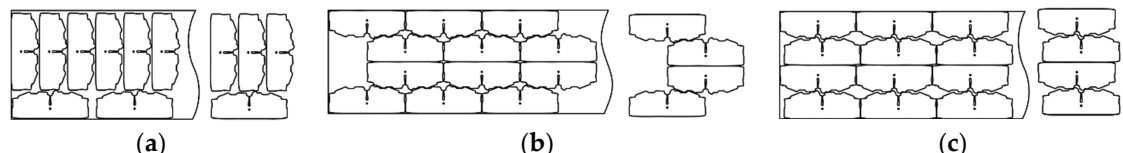

**Figure 12.** Part 3: (**a**) manual layout; (**b**) PSO; (**c**) unit nesting approach.

**Table 3.** Comparison of the results for a single part.

| Part | | Punching Manual Layout | Manual Layout | Laser Cutting PSO | Unit Nesting Approach |
|:---:|:---|:---:|:---:|:---:|:---:|
| | Step length (mm) | — | 901 | 901 | 901 |
| 1 | Utilization rate (%) | 61.06% | 66.12% | 88.16% | 88.16% |
| | Running time (s) | — | — | 2.37 | 1.98 |
| | Step length (mm) | — | 318 | 1886 | 1714 |
| 2 | Utilization rate (%) | 58.30% | 60.25% | 60.97% | 67.09% |
| | Running time (s) | — | — | 4.71 | 4.55 |
| | Step length (mm) | — | 775 | 707 | 707 |
| 3 | Utilization rate (%) | 75.13% | 81.72% | 90.60% | 90.60% |
| | Running time (s) | — | — | 3.04 | 2.58 |

### 4.2. Case 2

In Case 2, the nesting pattern of three mixed parts is required. Table 4 shows the information of three parts. Figure 13 is the optimized nesting pattern and unit of the different algorithms. Table 5 is a comparison of the step length and the utilization.

**Table 4.** Information of parts.

| Part Number | Legend | Size (mm × mm) | Coil Width (mm) |
|:---:|:---:|:---:|:---:|
| 1 | | 1115 × 878 | |
| 2 | | 1448 × 1363 | 2550 |
| 3 | | 1282 × 878 | |

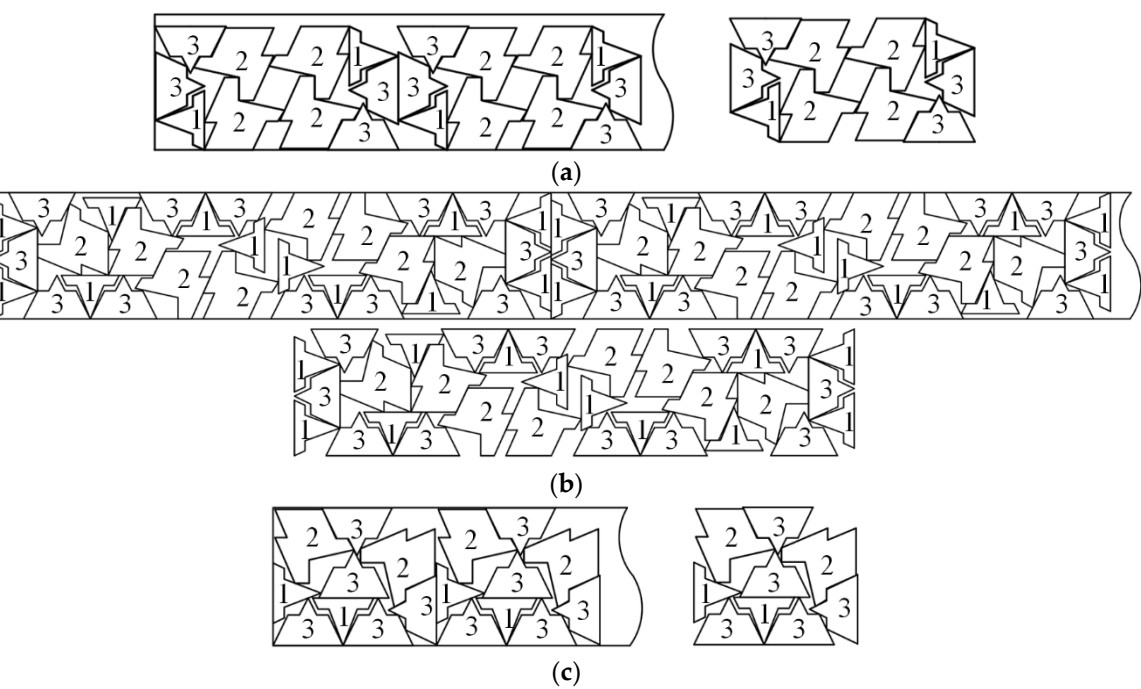

**Figure 13.** Diagram of different algorithms for layout: (**a**) manual layout; (**b**) PSO; (**c**) unit nesting approach (the numbers 1–3 in the figure are part numbering).

**Table 5.** Comparison of the results for mixed parts.

| Cutting Method | Layout Algorithm | Step Length (mm) | Utilization Rate (%) | Running Time (s) |
|:---:|:---:|:---:|:---:|:---:|
| Punching | Manual layout | — | 63.35% | — |
| Laser cutting | Manual layout | 4371 | 67.87% | — |
| | PSO | 10,763 | 75.28% | 14.27 |
| | Unit nesting approach | 2992 | 79.03% | 10.43 |

### 4.3. Results Analysis

From the data in Table 3: Compared with the manual nesting in laser cutting, the utilization rates of the three parts of the unit nesting approach are 22.04%, 6.84%, and 8.88% higher, respectively. Compared with the PSO, the utilization rates of part 1 and part 3 are the same, while the utilization rate of part 2 is 6.12% higher. The calculation times of the three parts in this algorithm are 0.39 s, 0.16 s, and 0.46 s faster than that of PSO, respectively.

From the data in Table 5: Compared with the manual nesting in laser cutting, the algorithm in this paper is 11.16% higher. Compared with the PSO, the utilization rate of the unit nesting approach is 3.75% higher and the calculation time is 3.84 s faster. Therefore, the unit nesting approach is feasible and effective in solving the problem of coil cutting.

## 5. Discussion

In this paper, a unit nesting approach is proposed for the cutting problem of coil material. It adjusts the sequence of parts by the improved marine predator algorithm and uses LH-TOPOS as the decoding method to obtain the unit with the maximum utilization rate. Compared with other methods, the unit nesting approach has the following advantages: (1) the nesting map is obtained by repeated movement of the unit, which can realize the requirements of the automatic continuous cutting production line; (2) compared with other algorithms, the marine predator algorithm can more easily jump out of the local optimal value. The computational results of Chapter 4 prove that the unit nesting approach not only increases the utilization to a certain extent but also greatly improves the calculation times.

The unit nesting algorithm solves the problems of a lack of continuity and low utilization in laser cutting to a certain extent; however, limitations exist. Firstly, the mathematical model is for a single target and will cause excessive production of some parts. Secondly, some parameters of the improved marine predator algorithm are set according to previous experience, which may affect the performance of the algorithm. Addressing these issues will be the direction of our future research.

**Author Contributions:** D.Q.: conceptualization, investigation, methodology, and writing—original draft preparation. W.Y.: methodology and writing—review and editing. L.D. and Y.W. (Yunzhi Wu): methodology and writing—review and editing. C.T. and L.Y.: writing—review and editing. Y.W. (Yuanfang Wang): methodology, writing—review and editing, supervision, and funding acquisition. Z.H.: resources, and funding acquisition. All authors have read and agreed to the published version of the manuscript.

**Funding:** This work was supported by the National Key Research and Development Program of China under Grant No. 2018YFD0700604.

**Institutional Review Board Statement:** Not applicable.

**Informed Consent Statement:** Not applicable.

**Data Availability Statement:** The data is in the article.

**Conflicts of Interest:** The authors declare that they have no known competing financial interest or personal relationships that could appear have to influenced the work reported in this paper.

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
