# Peer review of "An Intelligent Approach to the Unit Nesting Problem of Coil Material"

_applsci, doi:10.3390/app13169067_

Round 1
Reviewer 1 Report
All Figures are of poor quality and need improvement.
The paper starts by stating that the primary cutting of sheet metal parts has transitioned from punching to laser cutting, but it doesn't provide sufficient context or background information on this transition.
It would be helpful to provide a brief overview of the advantages and disadvantages of both methods and explain why laser cutting has become the main production method.
The text does not adequately review the existing literature on nesting methods for the continuous cutting of coil material. It is essential to establish the current state of the field and highlight the existing approaches, algorithms, or techniques that have been previously proposed and their limitations.
The paper briefly mentions that a unit nesting model is constructed and an improved marine predator algorithm is used to solve it. However, the details of the algorithm, its improvements over the standard marine predator algorithm, and how it addresses the specific challenges of coil material cutting are not adequately explained.
The paper claims that the presented approach is feasible and effective in improving material utilization and reducing production costs while meeting the requirements of the production site. However, there is no substantial evidence provided to support these claims. The paper should include a detailed experimental setup, performance metrics, and a comparison with existing methods to validate the effectiveness of the proposed approach.
The paper briefly mentions obtaining nesting units for production by the single row or opposite row of the minimum nesting unit, but it lacks a thorough discussion on the practical implementation of the proposed approach. Factors such as real-world constraints, scalability, and integration with existing manufacturing systems should be addressed to provide a comprehensive understanding of the applicability of the method.
The paper could benefit from better organization and clarity. Some sentences are ambiguous or overly complex, making it difficult to understand the intended meaning. Rewriting and proofreading the paper for grammar, syntax, and coherence would improve its readability.
Moderate editing of English language required
Author Response
Thank you for your kind comments on our manuscript entitled “An intelligent approach to unit nesting problem of coil material”. We have carefully revised the manuscript according to the reviewer’s comments. Based on the suggestions, we have made an extensive modification on the revised manuscript. Detailed revision was shown as follows. The changes to our manuscript within the document were also highlighted by using red colored text. The details of the reply to the question are in the attachment (response to reviewer 1).

Reviewer 2 Report
How is running time measured?
line 31: instead of "especial" use "specification of"
line 35: instead of "possible that laser" use "preferrable to use laser"
line 42: "problem."
line 51: " a new"
line 54,56,58: use only the surnames while giving references
line 62: instead of "successfully in" use "successfully implented to"
line 66,67: "material." "model." "Section 3."
line 70: "sheet material"
line 71: What "litter" refers to?
line 86: "sheet may not be fully applicable to" should be explained by giving any reason or references.
Along whole manuscript check for the dots before conjunctions.such as "while" and "where".
line 168: "Simulated Annealing ("
While giving constant parameter values such T and beta, how these values are selected should be explained.
line 237: "a system"
Table 3: why, for the third example, step length values is not given could be explained.
For all references, end of rows should be checked considering hyphenation rules.
Author Response
Thank you for your kind comments on our manuscript entitled “An intelligent approach to unit nesting problem of coil material”. We have carefully revised the manuscript according to the reviewer’s comments. Based on the suggestions, we have made an extensive modification on the revised manuscript. Detailed revision was shown as follows. The changes to our manuscript within the document were also highlighted by using red colored text. The details of the reply to the question are in the attachment (response to reviewer 2).

Reviewer 3 Report
The manuscript is written with poor language and almost no novelty.
There are many lingual errors in the paper throughout.
Author Response
Thank you for your kind comments on our manuscript entitled “An intelligent approach to unit nesting problem of coil material”. We have carefully revised the manuscript according to the reviewer’s comments. Based on the suggestions, we have made an extensive modification on the revised manuscript. Detailed revision was shown as follows. The changes to our manuscript within the document were also highlighted by using red colored text. The details of the reply to the question are in the attachment.

Reviewer 4 Report
the presented study entitled "An intelligent approach for unit nesting problem of coil material" is dedicated to the numerical solution of the optimization of the distribution of parts of a defined geometry during laser cutting in a coil of sheet metal. The authors use the predator-prey synergistic model as a basic starting point. In the introduction, the authors point out different approaches to the solution of the given problem based on the literature research. Here, however, the authors could also devote space to the methodology of the solved problem, and identify where the predator-prey models were applied, in which problems and with what result. Certainly such studies exist. At the same time, I lack an answer to the question of the authors' own scientific contribution in the introduction. So how is your approach and your study different, compared to the solutions of other authors? It would be necessary to expand the introduction with specific conclusions used in the study and to identify the research gap that would be the basis of the justification of the study. The model used to optimize the number of parts per roll for laser cutting is described relatively simply, without in-depth analysis. This reminder also applies to the results themselves. It would be interesting to simulate other dimensions, or shapes of the final product. At the same time, I would also welcome a practical verification of the approach to the solved problem. Theoretical analyzes are often at odds with real practice, so the calculations shown in tab. 5 are debatable. At the same time, I miss the validity limits of the given study, for which materials, shapes and dimensions of components, technological devices these results apply. The conclusion (which is absent in the study and is replaced by a discussion) is written too generally, which is also based on the study itself. Since the study by its nature belongs to the field of operational/production management, I recommend the authors to devote themselves to this aspect in more detail. Conclusion: the presented study needs to implement the modifications that I mentioned above, and therefore I do not recommend publishing the study in this form.Author Response
Thank you for your kind comments on our manuscript entitled “An intelligent approach to unit nesting problem of coil material”. We have carefully revised the manuscript according to the reviewer’s comments. Based on the suggestions, we have made an extensive modification on the revised manuscript. Detailed revision was shown as follows. The changes to our manuscript within the document were also highlighted by using red colored text. The details of the reply to the question are in the attachment.

Reviewer 5 Report
The article deals with an intelligent approach for unit nesting problem of coil material. Overall, the article is good. In the area of improvement, I suggest: moving the images from the Introduction to the Materials and Methods section. Increase the clarity and specificity of the introduction by explicitly defining the objectives and key questions your study addresses, so readers better understand the main focus and goal of your work. Improve the quality of Figures 3, 7, and 8. Define and summarize the main findings and contributions of your work more clearly. Competed what does your research add to existing knowledge, and why is this important. Provide more specific details about how your method "improves utilization to some extent and significantly increases efficiency". Add which aspects of efficiency are improved and euphemize quantitative data that would support this. Emphasize and explain more thoroughly the limitations of your research and how these limitations affect the interpretation and application of the results. Suggest more specific directions for future research.
Author Response

(The authors gave the same response as above.)

Reviewer 6 Report
The authors have used an improved marine predator algorithm for nesting optimization problems for an irregular part for continuous cutting of coil material to improve productivity.
List the assumptions followed I this method.
In section 2.1, the formula applies for a searching space for sequential operations, if there exist parallel operations the space may increase / decrease.
Improve the quality of figures.
Represent the sub/superscripts properly for the symbols listed in Table 1.
The objective function is generally to minimize the material utilisation, which is missing in this case.
There exist numerous nesting optimisation tools associated with sheet metal design and laminate composite design, plesde justify the efficiency of the proposed frame work. Refer the below literature.
Jiang, Chao, Xu Han, and G. P. Liu. "Uncertain optimization of composite laminated plates using a nonlinear interval number programming method." Computers & Structures 86.17-18 (2008): 1696-1703.
Prasad, VSS Vara, et al. "A novel computative strategic planning projections algorithm (CSPPA) to generate oblique directional interference matrix for different applications in computer-aided design." Computers in Industry 141 (2022): 103703.
Diyaley, Sunny, and Shankar Chakraborty. "Metaheuristics-based nesting of parts in sheet metal cutting operation." Operational Research in Engineering Sciences: Theory and Applications 5.2 (2022): 1-16.
Please do a comparative assessment with recent literature to draw the merits of the proposed framework.
Author Response
Thank you for your kind comments on our manuscript entitled “An intelligent approach to unit nesting problem of coil material”. We have carefully revised the manuscript according to the reviewer’s comments. Based on the suggestions, we have made an extensive modification on the revised manuscript. Detailed revision was shown as follows. The changes to our manuscript within the document were also highlighted by using red colored text. The details of the reply to the question are in the attachment .

Round 2
Reviewer 1 Report
Accept
Reviewer 4 Report
After supplementing and reworking the study entitled An intelligent approach for unit nesting problem of coil material, auti significantly improved its quality. The authors incorporated my comments into the manuscript and at the same time additionally explained some disputed points. Therefore, after re-examining the manuscript, I have no further comments and recommend publishing the study.